# Pixel Co-Occurence Based Loss Metrics for Super Resolution Texture Recovery

## Abstract

Single Image Super Resolution (SISR) has significantly improved with Convolutional Neural Networks (CNNs) and Generative Adversarial Networks (GANs), often achieving order of magnitude better pixelwise accuracies (distortions) and state-of-the-art perceptual accuracy. Due to the stochastic nature of GAN reconstruction and the ill-posed nature of the problem, perceptual accuracy tends to correlate inversely with pixelwise accuracy which is especially detrimental to SISR, where preservation of original content is an objective. GAN stochastics can be guided by intermediate loss functions such as the VGG featurewise loss, but these features are typically derived from biased pre-trained networks. Similarly, measurements of perceptual quality such as the human Mean Opinion Score (MOS) and no-reference measures have issues with pre-trained bias. The spatial relationships between pixel values can be measured without bias using the Grey Level Co-occurence Matrix (GLCM), which was found to match the cardinality and comparative value of the MOS while reducing subjectivity and automating the analytical process. In this work, the GLCM is also directly used as a loss function to guide the generation of perceptually accurate images based on spatial collocation of pixel values. We compare GLCM based loss against scenarios where (1) no intermediate guiding loss function, and (2) the VGG feature function are used. Experimental validation is carried on X-ray images of rock samples, characterised by significant number of high frequency texture features. We find GLCM-based loss to result in images with higher pixelwise accuracy and better perceptual scores.

## 1 Introduction

A super resolution (SR) image is generated from a single low resolution image (LR) (with or without variable blur and noise) such that the result closely matches the true high resolution counterpart (whether it exists or not) (Park et al., 2003). There thus exists a vast number of possible solutions (Dong et al., 2014) for any given LR image, and by an extension, there are many techniques to recover SR details with varying degrees of accuracy. These methods range from the simple and blurry interpolation methods (bicubic, linear, etc.) that cannot recover contextual features to more complex model-based methods that utilise prior knowledge of certain domain characteristics to generate sharper details (Dai et al., 2009; Jian Sun et al., 2008; Yan et al., 2015). The contextual dependence of SR images is addressed with Example-based approaches including Markov Random Field (MRF) (Freeman et al., 2002), Neighbour Embedding (Hong Chang et al., 2004), Sparse Coding (Yang et al., 2010) and Random Forest methods (Schulter et al., 2015). These can be generalised (Dong et al., 2014) into Super Resolution Convolutional Neural Networks (SRCNN) for photographic images (Dong et al., 2014; Wang et al., 2016; Lim et al., 2017; Yu et al., 2018; Kim et al., 2015; Ledig et al., 2017), medical images (Umehara et al., 2018; You et al., 2018), and digital rock images (Wang et al., 2019b; 2018; 2019a). The original SRCNN network utilised as a backbone 3-5 convolutional layers (Dong et al., 2014) that sharpened a bicubically upsampled image. More recent models contain dedicated upsampling layers (Dong et al., 2016; Shi et al., 2016; Odena et al., 2016), skip connections to improve gradient scaling (Kim et al., 2015; Ledig et al., 2017), removal of batch normalisation (Lim et al., 2017), and use of the sharper L1 loss function (Yu et al., 2018; Zhu et al., 2017). These changes have all contributed gradually to improving the pixelwise accuracy of the super resolution

methods. Despite the high pixelwise accuracy achieved by SRCNN networks [1], the resulting super resolved images are often perceptually unsatisfying, and easily identifiable by a human observer as "blurry". It is due to the SRCNN can recover accurately features spanning several pixels such as larger scale edges, texture and high frequency features are lost as the network attempts to maximise the pixelwise accuracy over a wide range of possible HR counterparts. The local minima problem thus manifests itself clearly in the ill-posed problem of Super Resolution.

Perceptual results can be improved by training the network with a hybrid loss function that combines a pixelwise loss (L1 or L2) with a feature-wise loss that is the L2 loss of features extracted from some intermediate convolutional layer of a pre-trained model (Johnson et al., 2016). The most effective method of perceptual texture generation thus far has been the use of Generative Adversarial Networks (GANs) (Goodfellow et al., 2014), with SRGAN generated images from low resolution photographs score highly on human surveys of image quality (Ledig et al., 2017; Dosovitskiy & Brox, 2016) as they can recover high frequency textures that are perceived as realistic, both for photographic features (Ledig et al., 2017) and textural quality in X-ray images (You et al., 2018). However, SRGAN generated images are stochastic in the way that high frequency features are regenerated, with a tendency to cause pixel mismatch, which is especially exacerbated by the use of the VGG feature loss (You et al., 2018; Ledig et al., 2017; Sajjadi et al., 2017) that further tends to result in further distortion, leading to higher pixelwise loss. For natural images this loss in pixelwise accuracy may be secondary, but plays an important role in applications where texture carries actual information, such as in X-ray images. This trade-off between the pixelwise accuracy (distortion) and the perceptual accuracy (Sajjadi et al., 2017; Mechrez et al., 2018; Vasu et al., 2018), is a consistently emergent limitation in SRGAN performance, whereby a high pixelwise accuracy causes over-smoothing, while a high perceptual accuracy causes pixel mismatch and distortion in some features. While both pixelwise accuracy and perceptual accuracy are important, SISR aims to preserve as much content/characteristics from the original image, while GAN essentially "makes up" content that is perceptually satisfying at the expense of pixelwise measures.

While pixel-wise distortions can be quantitatively measured, in order to evaluate perceptual performance one typically requires human subjective evaluations of Mean Opinion Scores (MOS), or by leveraging the proxy score produced by "pre-trained" models (which could introduce their own biases) (Ma et al., 2016; Salimans et al., 2016). SRGANs typically obtain superior scores in terms of subjective metrics (Ledig et al., 2017; Zhu et al., 2017; Vasu et al., 2018), however, an objective study of the differences in the high frequency SR textures compared to the original HR images has not yet been carried. This is especially of interest in imaging areas requiring expert judgements, such as in radiographic images. These are most often benchmarked with a combination of pixelwise metrics and opinion score surveys despite the smaller sample sizes and logistical challenge.

The overall aim of this study is the introduction of Grey Level Co-occurence Matrix (GLCM) method (Haralick et al., 1973) as both an auxiliary loss function, and as an addition to the PSNR metric for evaluating perceptual and textural accuracy of super resolution techniques (GLCM better correlates with the subjective scores of human evaluations). GLCM has been sucessfully used in the characterisation of texture in medical images (Makaju et al., 2018; Madero Orozco et al., 2015; Sivaramakrishna et al., 2002; Pratiwi et al., 2015; Liao et al., 2011; Yang et al., 2012) and CT rock images (Singh et al., 2019; Becker et al., 2016; Jardine et al., 2018). In essence, the GLCM transforms an image into a representation of the spatial relationship between the grey colours present in the image. The GLCM is not a pixelwise measurement, and so can be used to evaluate texture recovery that may not be a pixelwise match. The GLCM is particularly well suited for automatic perceptual/textural in-domain comparisons, as it does not require time-consuming expert MOS evaluations, and does not pose inherent data-bias when scoring with auxiliary models.

We use the DeepRock-SR (Wang et al., 2019a) dataset to train and validate the results of an SRGAN model, which in this study is a modified Enhanced Deep Super Resolution (EDSR) network (Lim et al., 2017) coupled to a GAN discriminator (EDSRGAN). The resulting performance in generating SR images is quantitatively analysed, both based on the traditional pixelwise approach, as well as using the GLCM spatial texture accuracy method introduced in this work. GLCM texture analyses of the SRGAN and SRCNN results indicate quantitatively that SRGAN produces images with a GLCM error that are an order of magnitude lower (more texturally similar) than SRCNN images, that tend to

---

[1]Henceforth, unless stated otherwise, all generative convolutional networks (and the corresponding generated images) will be referred to as "SRCNN"

have a higher PSNR, but also a higher GLCM error and lower MOS scores. Overall, the use of the GLCM offers fast and more agnostic evaluation metric when compared to carrying MOS evaluations, and is easier to reproduce and analyse due to its algorithmic data-driven nature. The GLCM can be also used as an auxiliary loss function to guide the generation of spatially accurate texture, resulting in further reductions in the texture error while also improving the pixelwise accuracy.

## 2 METHODS

### 2.1 GREY LEVEL CO-OCCURRENCE MATRIX FOR TEXTURE ANALYSIS

The Grey Level Co-occurrence Matrix characterises the texture of an image by calculating the number of pairs of pixels that could be spatially related within some pre-defined regions of interest. For a given image with $N$ grey levels, an $N$ by $N$ co-occurrence matrix $P$ is constructed, in which the $(i, j)$ position denotes the number of times a grey level $i$ is spatially related to a corresponding grey level $j$. An example of spatial relationship setting in an image of size $(N_x, N_y)$ could be all locations of $(x, y)$ and $(x + 1, y)$ (i.e. their horizontal adjacency). In this case, the value at location $(i, j)$ in $P$ is the sum of all occurrences where grey level $i$ and grey level $j$ occur horizontally adjacent to each other within the image. Since the GLCM does not compare spatially matching pixel values but instead compares the spatial distribution of pixel values, it is a good measure of texture similarity to complement the pixel-by-pixel similarity PSNR metric.

In general, multiple GLCMs with an encompassing set of spatial relationships are constructed to fully characterise the texture of an image. Aside from the default $(x, y)$ to $(x + 1, y)$ relationship, the offset value can be generalised to an omnidirectional pixel distance $d \in \mathbb{Z}$ such that the spatial relationships are $(x, y)$ to $(x + d_x, y + d_y)$. Some of the the GLCMs used in this study are calculated in 8 directions with a 45 degree offset, to a distance of up to 10 pixels with a 4-bit precision (16 grey levels after quantisation). This results in a 8x10x16x16 GLCM tensor $\overline{\overline{P}}$, or 80 16x16 GLCMS, one for each spatial relationship setting. This raw transformation can then be analysed for a variety of statistical measures, or a pixelwise comparison can be carried on GLCMs for different images to comparatively quantify texture similarity. If a generated SR image is texturally similar to the original HR image, then it can be expected that the corresponding GLCMs are closer to each other w.r.t L1 or L2 distances. The L1-GLCM error is computed as:

$$GLCM_{Loss} = \frac{\sum_{ij} |\overline{\overline{P}}_{SR_{ij}} - \overline{\overline{P}}_{HR_{ij}}|}{\sum_{ij} ij} \tag{1}$$

### 2.2 SUPER RESOLUTION GENERATIVE ADVERSARIAL NETWORK AND LOSS FUNCTIONS

The architecture used in this study is the EDSRGAN network (Wang et al., 2019a), which is based on the Enhanced Deep Super Resolution (EDSR) (Lim et al., 2017) and the SR-Resnet (Ledig et al., 2017) networks. The model uses Parametric Rectified Linear Units (PReLU) as activation functions, and batch normalisation layers are removed as this was found detrimental to SR convergence (Lim et al., 2017; Yu et al., 2018). The Generator $G$ is depicted in Figure 1. The L1 pixelwise loss was used as it offered better SRCNN convergence and less severe smearing compared to the L2 loss (Zhu et al., 2017) by reducing the penalty for high frequency noise and texture.

$G$ is trained for 100,000 iterations on cropped mini-batches of 16x192x192 with a learning rate of 1e-4 using the Adam optimiser (Kingma & Ba, 2014). The generator $G$ is capable of boosting image resolution and sharpening medium to large scale features (Wang et al., 2019b;a). SRGAN is finetuned to regenerate high frequency textures and features smaller than 3-5 pixels. The GAN couples $G$ with a discriminator $D$ depicted in Figure 2 by the adversarial loss $ADV_{Loss}$. Joint training continues for additional 150,000 iterations.

On top of $L_{1_{Loss}}$ and $ADV_{Loss}$ losses, the total generator loss also includes VGG-19 perceptual objective (Ledig et al., 2017), and the proposed $GLCM_{Loss}$ (cf. equation 1), and is defined as follows:

$$G_{Loss} = L_{1_{Loss}} + \alpha VGG_{19_{Loss}} + \beta ADV_{Loss} + \gamma GLCM_{Loss} \tag{2}$$

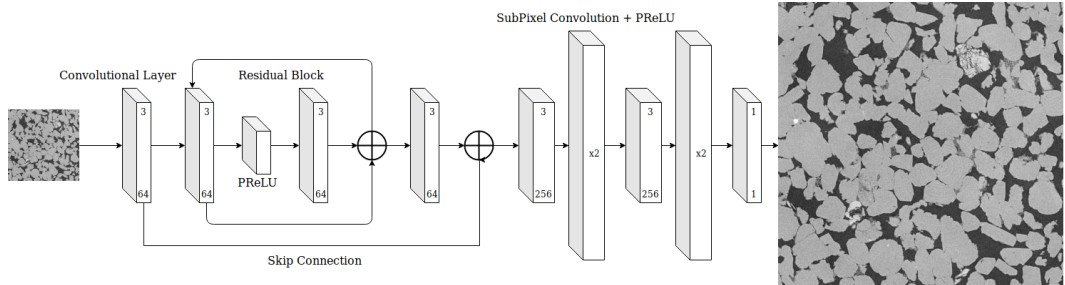

Figure 1: Architecture of the generator network. As in EDSR network but with PReLU activations in all layers, including subpixel convolutions during upscaling.

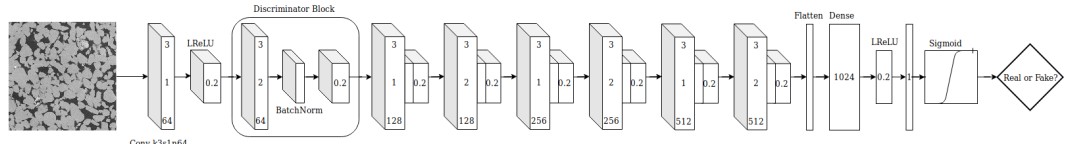

Figure 2: Architecture of the Discriminator network.

where $\alpha$, $\beta$, and $\gamma$ are scaling terms set to 1e-5, 1e-3, and 1e-4, respectively to keep their magnitudes in the ballpark of the L1 loss.

## 3 RESULTS AND DISCUSSION

This section shows the results and analyses of the EDSRGAN network performance, when trained with GLCM metric. The first set of experiments illustrate GLCM performance as a metric to quantify texture accuracy of the network in its standard configuration with the VGG19 loss function. Then setup gets extended to using GLCM as an auxiliary loss function as shown in equation 2.

### 3.1 GREY LEVEL CO-OCCURENCE MATRIX AS A METRIC

The pixelwise accuracy of the SRCNN and SRGAN is calculated by comparing the PSNR of the Bicubic (BC), SR, and SRGAN images against the ground truth HR images. The PSNR results are reported on a held-out set of 800 500x500 test images of sandstone, carbonate, and coal in Table 1. In terms of pixelwise accuracy, SRGAN images are less accurate compared to SR images. This is expected for perceptual losses, as they sample one of many plausible SR solutions, of which HR is only one instance.

Table 1: PSNR results for the DeepRock-SR dataset (Higher is better)

|  | Sandstone |  | Carbonate |  | Coal |  |
|---|---|---|---|---|---|---|
| PSNR (dB) | Mean | Var | Mean | Var | Mean | Var |
| Bicubic | 25.8822 | 0.6802 | 22.9768 | 1.9896 | 39.8738 | 10.3738 |
| SRCNN | 28.5986 | 6.1992 | 24.3879 | 1.7475 | 42.6653 | 3.7016 |
| SRGAN | 26.2118 | 8.3945 | 21.8533 | 1.3415 | 40.6061 | 4.4339 |

Sample visualisations of the resulting validation images are shown in Figure 3. The HR image (top row) and LR image (first column), and the corresponding SR variants obtained with bicubic interpolation (BC), SRCNN and SRGAN methods (3rd - 5th column, respectively). A trend of gradually improved visual sharpness and texture quality from BC to SR and SR to SRGAN images can be observed. This is particularly clear for carbonate images that contain highly heterogeneous features such as oolitic vugs and high frequency texture associated with microporosity. This visual trend is as expected from SRGAN type networks, but is entirely lost when looking at the PSNR results in Table 1. In other words, the pixel mismatch caused by stochastic GAN generation is unaccounted for, causing the drop in the pixelwise accuracy of SRGAN images. Thus, the spatial relationship of

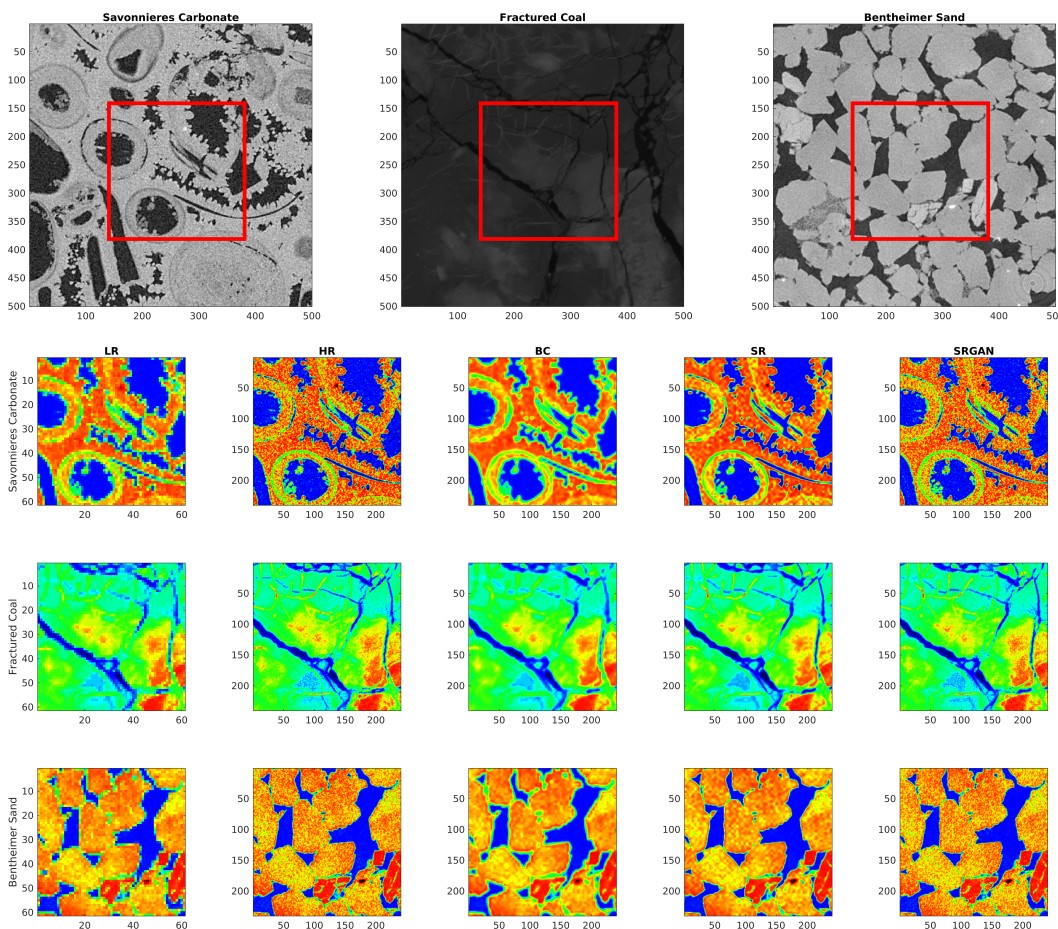

*Figure 3: Sample images from the validation set using EDSRGAN network. Based on the LR image, a bicubic SR, SRCNN and SRGAN images are generated. Results show improvement in feature recovery in SRCNN images, and texture recovery in SRGAN images that look perceptually identical to the HR images.*

pixels is analysed instead of the pixelwise relationship. To do this, GLCMs are calculated for HR, SR, SRGAN, and BC images and compared to each other. As an initial reference, the 8-way 4-bit GLCM over an offset of 10 pixels (see Section 2.1) is used. The L1 error between the GLCMs is calculated as a measure of textural similarity and plotted for the validation images, presented in Figure 4 sorted in ascending order of L1 GLCM error.

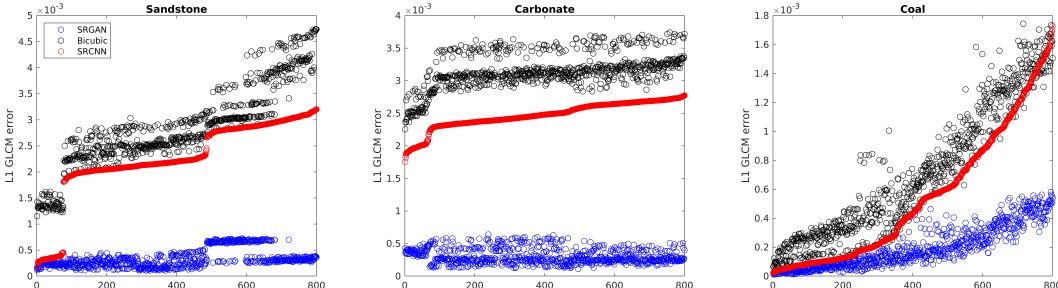

*Figure 4: Plots of the L1 GLCM error for sandstone, carbonate, and coal images in the validation and testing sets of the DeepRock-SR dataset. 800 images for each rock type are sorted in ascending order of SRGAN error for visual clarity. It can be seen that SRGAN results show a clear comparative improvement in spatial texture similarity.*

There is a clear delineation between the BC, SR, and SRGAN images in terms of their L1-GLCM error, with SRGAN images offering best results in terms of their GLCM defined spatial relationships. The filtered Gildehauser sandstone images can be seen as the red data points close to the blue SRGAN line in the sandstone graph, and show lower overall error compared to the other raw sandstone images. Another sample of interest is the Wilcox Tight Gas Sandstone, seen as the outlying cloud of points that results in high comparative GLCM errors in Bicubic and SRCNN methods due to the increase in intragranular texture compared to conventional sandstones. Carbonate images show similar improvements in texture similarity. Coal images possess a sparse distribution of fracture features within an otherwise fairly smooth sub resolution micro-fracture that results in low proportion of regions with high texture variation. Thus the SRCNN and bicubic methods show a lesser degree of texture loss compared to SRGAN. While SRCNN possesses a greater pixel-by-pixel accuracy, the texture error analysis in this section shows that SRGAN is superior in recovering the spatial relationship between pixel values. Since this 8-way 4-bit 10px offset choice of GLCM is arbitrary, whether this delineation of texture accuracy is preserved when using different GLCM parameters is shown in Figure 5.

Changing the bit depth results in quantisation of the spatial grey values, and causes a loss of information which can smear away certain details. Changing the offset length affects the weighting and importance assigned to the spatial relationship of pixels close (i.e. 5px) and far (i.e 20px) neighbourhoods. We find that, when plotting the median L1-GLCM error for each rock type over different bit depths and offset lengths, the relationship between order of texture accuracy as compared to the HR ground truth image is preserved without exception. This lends credence to the general use of the L1-GLCM as a measurement of perceptual error in textured images.

## 3.2 GREY LEVEL CO-OCCURENCE MATRIX AS A LOSS FUNCTION

Since the GLCM metric clearly quantifies and delineates texture accuracy, it can be directly used as a loss function to replace the VGG loss function that is canonically used in the SRGAN network. In this case, the GLCM is calculated with floating point precision to maintain differentiability, and to preserve computational speed during training, the GLCM is calculated with an offset of 2px in 8 directions. This 8-way, float32-bit GLCM with offset of 2px replaces the VGG function during training, and the resulting pixelwise and texturewise accuracy is compared against SRGAN-MSE (trained with no intermediate loss function), as well as SRGAN-VGG. Furthermore, the 8-way, 4-bit GLCM metric used in the previous section shows that the overall texture accuracy is also improved compared to SRGAN-VGG, as shown in Figure 6. The overall pixelwise and texturewise accuracy of SRGAN-GLCM is superior to versions with MSE-only and VGG as loss functions. This shows an ability to directly use the GLCM as an unbiased loss function is a further benefit aside from its potential use as a quality metric for texture recovery.

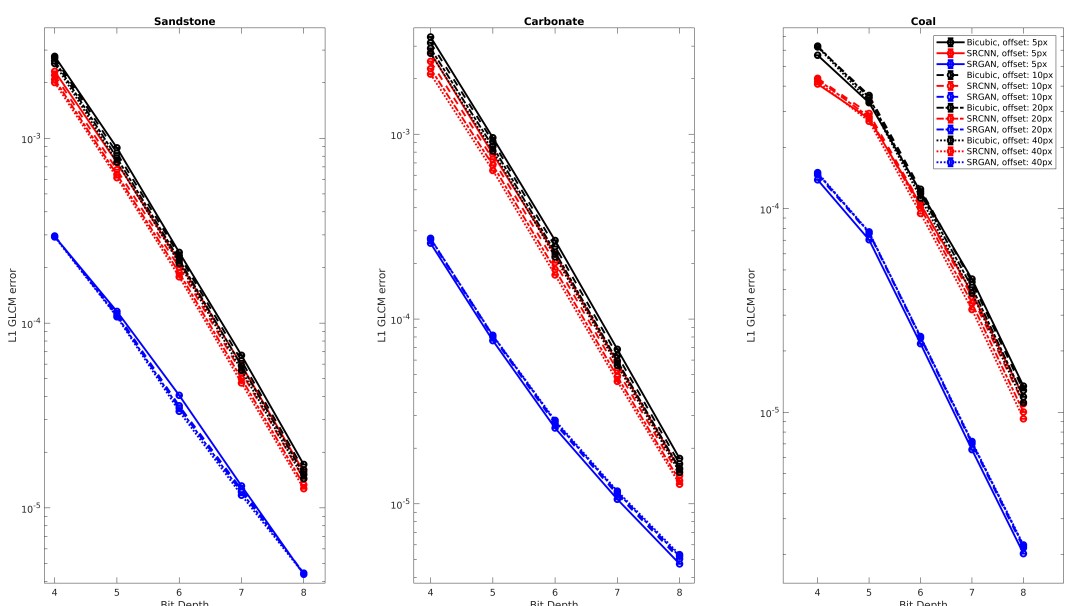

*Figure 5: Different bit depths and offset lengths for the L1 GLCM for sandstone, carbonate, and coal images retain the same cardinal relationship in terms of spatial texture reconstruction accuracy*

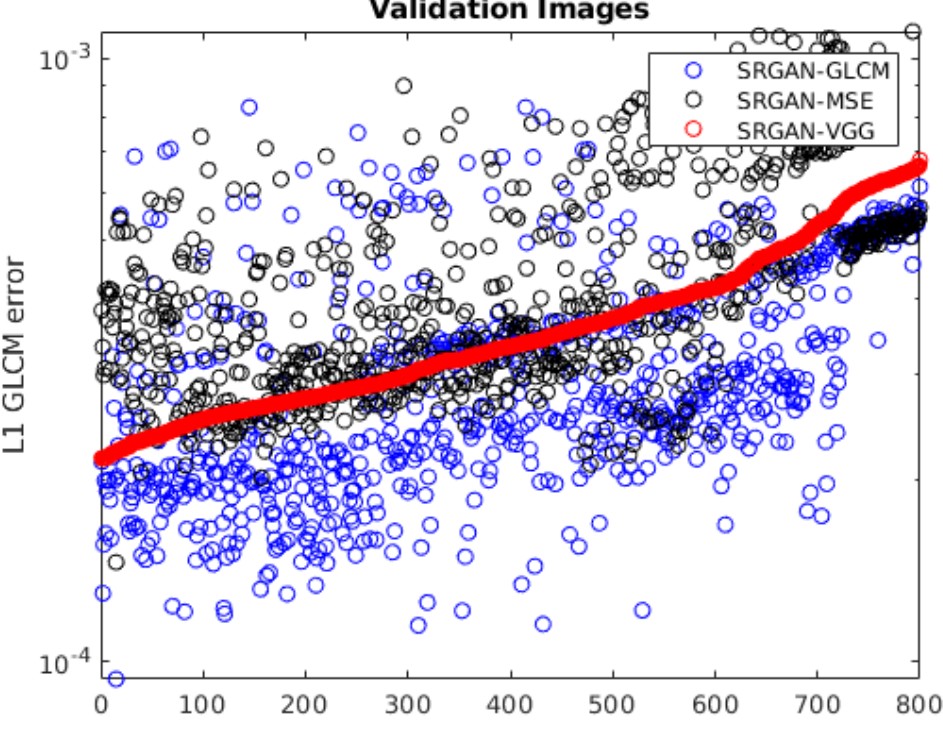

*Figure 6: Comparison of texture accuracy as measured by the L1-GLCM, for cases of SRGAN trained with MSE-only, VGG, and GLCM loss functions. The GLCM loss function produces the lowest overall texture error.*

The resulting trained networks produce a mean pixelwise and texturewise error as shown in Table 2, indicating that SRGAN-GLCM provides the most accurate overall result. The SRGAN-MSE results are obtained without any intermediate loss function that would guide the generation of SR images, while the SRGAN-VGG results make use of the VGG loss function. Note, in our case auxiliary VGG model was trained to detect coloured photographic features rather than greyscale X-ray textures. Thus the degree of pixelwise distortion is highest for SRGAN-VGG. SRGAN-MSE results are characterised by the highest texturewise error, since the GAN guided generation of perceptually accurate features is entirely stochastic and unguided. These limitations of SRGAN-MSE and SRGAN-VGG are less severe in the SRGAN-GLCM, which does not cause extra distortion of the pixelwise configuration (resulting in a higher PSNR). At the same time, SRGAN-GLCM also improves the spatial accuracy of pixel values, thus improving also texture accuracy.

*Table 2: Pixelwise (Higher is better) and texturewise (Lower is better) results for the DeepRock-SR dataset, comparing SRGAN network performance as trained using only the MSE, MSE+VGG, and MSE+GLCM. Average values on validation dataset, the pixelwise mismatch is highest for VGG loss, while lowest for GLCM loss. Texturewise, the GLCM is superior followed by VGG and MSE metrics.*

| Validation Images | | |
|---|---|---|
| | Mean PSNR (dB) | Mean L1-GLCM |
| SRGAN-MSE | 29.3787 | 4.465e-4 |
| SRGAN-VGG | 29.3344 | 3.651e-4 |
| SRGAN-GLCM | 29.3815 | 3.124e-4 |

## 4 CONCLUSIONS

The issue with understanding and quantifying the accuracy of perceptual texture regeneration is addressed in an unbiased and automated approach that uses the GLCM as an accuracy metric as well as a loss function. Texture analysis using GLCMs (of varying bit depths and offsets) shows superior texture similarity of SRGAN images compared to normal SRCNN and other interpolation methods. Similarly, using the GLCM as a loss function reduces distortions caused by the VGG loss function, which improves the pixelwise accuracy and reduces texturewise errors. GLCM, used either as metric and/or loss function, applied in this work to digital rock images, is likely to also perform well for other types of highly textured greyscale images such as medical images (where the concept of the GLCM originated from).

While the GLCM has been shown to act well with SRGAN networks to obtain higher pixelwise and texturewise accuracies on X-ray images, its performance both as the perceptual metric and loss function remains to be explored for natural photographic images. One would reasonably expect, that the GLCM would confer some (as yet untested) benefit for any vision generation tasks where high quality high frequency textures are present. Furthermore, the GLCM is traditionally used for greyscale images, so its use in colour images would require some modifications, such as a separate GLCM per colour channel. We leave this as a future work.

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

## 5 APPENDIX

### 5.1 TEXTURED X-RAY IMAGE DATASET

X-ray Computed Tomography (CT) images that detail the finer features within the internal structure of porous media are important in characterisation of the physical properties in material analysis (Schlüter et al., 2014), as well as improving medical diagnostic accuracy and radiomic analysis (Park et al., 2003; Yeh et al., 2010; Mathieson et al., 1989). One of the recent application is in earth science and engineering for determination of rock properties. Rock images, also known as digital rock, at higher energies and micrometre resolution ($\mu$CT) assist in determination of petrophysical and flow properties of rocks (Wang et al., 2019d; Chung et al., 2018; Wang et al., 2019c; Mostaghimi et al., 2013; Krakowska et al., 2016; Blunt et al., 2013) in a non-destructive manner (Lindquist et al., 1996; Hazlett, 1995; Wildenschild & Sheppard, 2013; Schlüter et al., 2014). A high resolution rock image that resolves pore space features while spanning a wide FOV to represent bulk properties (Li et al.,

2017) can be challenging to obtain in such cases, but can be generated by super resolution methods (Wang et al., 2019b;a).

The DeepRock-SR dataset comprises of 12,000 500x500 high resolution unsegmented slices of various digital rocks of sandstone, carbonate, and coal (4,000 images respectively), with image resolution ranging from 2.7 to 25 $\mu$m, outlined in previous studies (Wang et al., 2019a). The dataset contains a wide range of rock types, imaged under various conditions. Resolved sandstone grains and under-resolved clay minerals, complex under-resolved carbonate microporosity (Bultreys et al., 2015), and coal fracture networks (Hao Chen et al., 2014) are all encompassed as features within the dataset.

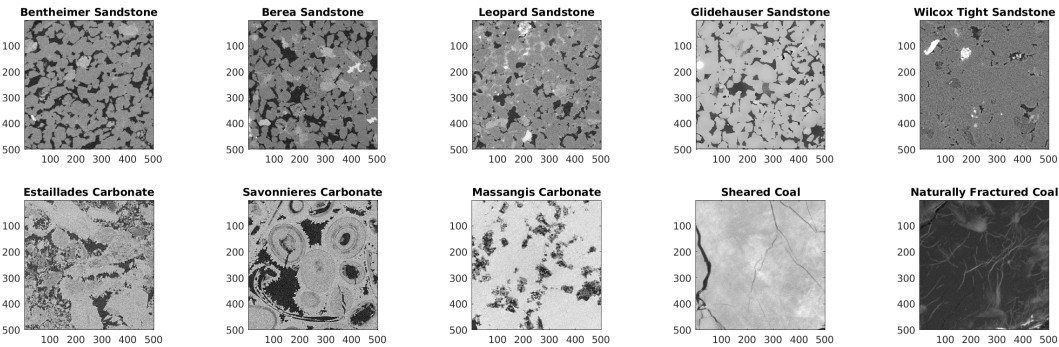

Figure 7: Sample slices of images from DeepRock-SR

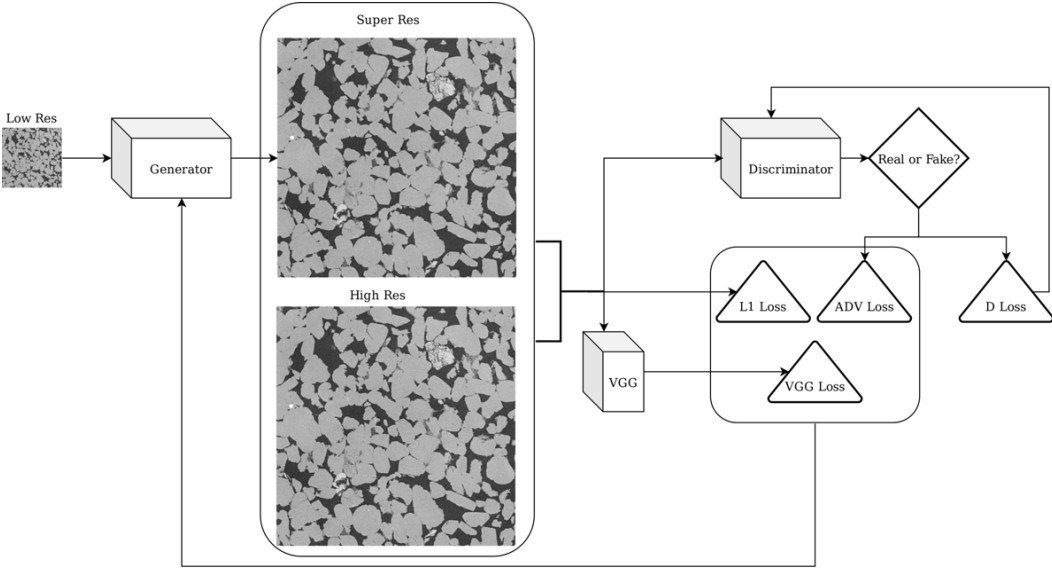

Figure 8: Overall architecture of the EDSRGAN network

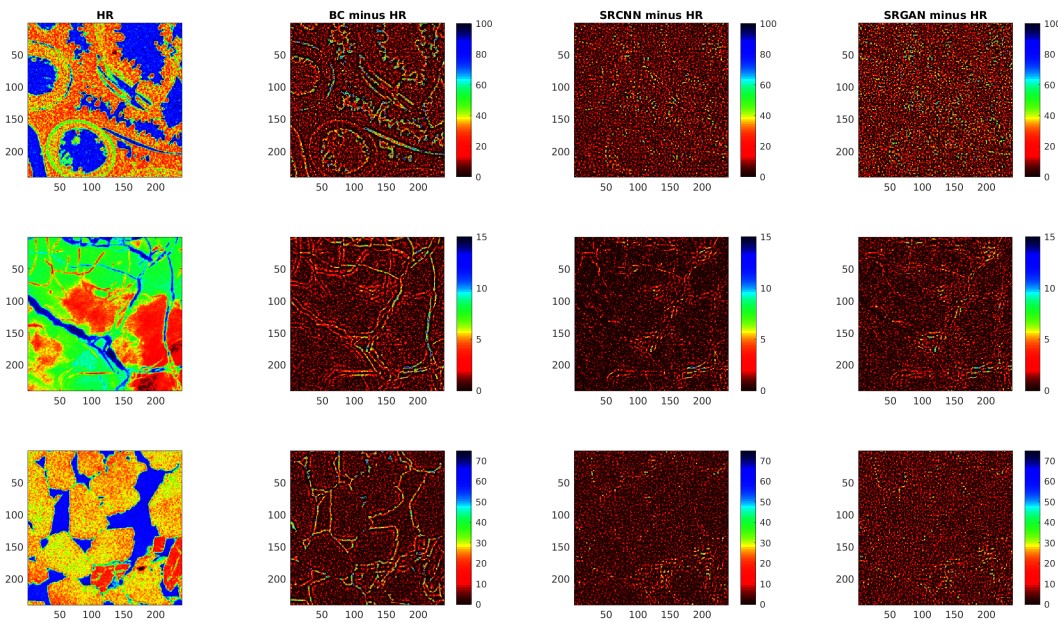

*Figure 9: Difference maps of the sample images from Figure 3, revealing no obvious improvement in the pixelwise accuracy of SRGAN images despite there being a high perceptual fidelity to the SRGAN images*

