# OpenReview forum: "Pixel Co-Occurence Based Loss Metrics for Super Resolution Texture Recovery"
_ICLR.cc/2020/Conference — Reject_

### Official Review · AnonReviewer2 · 2019-10-20
**Official Blind Review #2**

**Rating:** 1

**Review:**

This paper addresses the super-resolution problem. The key is to use pixel co-occurrence-based loss metric. The idea is very straightforward. But the description could be clearer. For example, what is the spatial size of P (\bar P)? How does it influence the optimization?

Equation (2): There are four loss functions on the right hand. How are the loss defined?

How is the GAN used?

In experiments, there is no evidence showing the benefit from the pixel Co-occurrence

There is a lack of much details. Given the current presentation, I cannot judge if the quality reaches the ICLR bar.

**Experience Assessment:**

I do not know much about this area.

**Review Assessment: Checking Correctness Of Derivations And Theory:**

N/A

**Review Assessment: Checking Correctness Of Experiments:**

I assessed the sensibility of the experiments.

**Review Assessment: Thoroughness In Paper Reading:**

I read the paper at least twice and used my best judgement in assessing the paper.

---

### Official Review · AnonReviewer1 · 2019-10-22
**Official Blind Review #1**

**Rating:** 1

**Review:**

The paper considers the problem of generating a high-resolution image from a low-resolution one. The paper introduces the Grey Level Co-occurrence Matrix Method (GLCM) for evaluating the performance of super-resolution techniques and as an auxiliary loss function for training neural networks to perform well for super-resolution. The GLCM was originally introduced in a 1973 paper and has been used in a few papers in the computer vision community. The paper trains and validates a super-resolution GAN (SRGAN)  and a super-resolution CNN (SRCNN) on the DeepRock-SR dataset. Specifically, for the SRGAN, the paper uses the EDSRGAN network trained on loss function particular to the paper: The loss function consists of the addition of the L1 pixel-wise loss plus the VGG19 perceptual objective plus the proposed GLCM loss.
The paper finds that SRCNN outperforms SRGAN in terms of the PSNR metric, but SRGAN performs better in terms of the spatial texture similarity.
Next, the paper shows that when trained with the mean L1-GMCM loss function, SRGAN performs best.

In summary, the paper proposes to use the GMCM loss for training and evaluation of the super-resolution methods. However, this metric is well known in the computer-vision community along with many others. Also, the idea to use this metric in training is only evaluated for one network (albeit a very sensible one) and only for one dataset (the DeepRock one). Since the novelty provided by the paper is small, I cannot recommend the acceptance of the paper.

**Experience Assessment:**

I have read many papers in this area.

**Review Assessment: Checking Correctness Of Derivations And Theory:**

N/A

**Review Assessment: Checking Correctness Of Experiments:**

I assessed the sensibility of the experiments.

**Review Assessment: Thoroughness In Paper Reading:**

I read the paper at least twice and used my best judgement in assessing the paper.

---

### Official Review · AnonReviewer3 · 2019-10-23
**Official Blind Review #3**

**Rating:** 3

**Review:**

This paper adopts a loss metric called Grey Level Co-occurence Matrix (GLCM) as a new measurement of perceptual quality for single image super-resolution. The GLCM is particularly well suited for automatic perceptual/textural in-domain comparisons, which does not require time-consuming expert MOS evaluations. Experimental validation is carried on X-ray images of rock samples and promising results are achieved.

My main concerns are as follows:
- Novelty is quite limited. First, as the main contribution of this work, GLCM is proposed from Haralick et al.(1973), which is not novel. Second, the network structure used in this work is also based on SRGAN/EDSRGAN, which is also not novel.

- Experiments are not convincing. In super-resolution task, the differences in Tab. 2 are quite minor, which can also be regarded as the same or may be caused by randomness. Moreover, the authors should conduct experiments on generic image SR to demonstrate the effectiveness of GLCM. Further, more qualitative visualizations are also needed to demonstrate the effectiveness of GLCM.

- Page 5 only contains 1 figure, leaving a lot of  space that is not fully used. Besides, Fig. 3 cannot reflect the advantage of GLCM, since SRGAN is much larger and also better than SRCNN.

**Experience Assessment:**

I have published in this field for several years.

**Review Assessment: Checking Correctness Of Derivations And Theory:**

N/A

**Review Assessment: Checking Correctness Of Experiments:**

I carefully checked the experiments.

**Review Assessment: Thoroughness In Paper Reading:**

I read the paper at least twice and used my best judgement in assessing the paper.

---

> ### Author Response · Authors · 2019-11-10
> **Novelty and Experiments**
>
> Regarding the comments:
>
> > We agree that the method we propose should be further tested on other image types, such as medical images and natural images. These experiments are in progress
>
> > The novelty of this work is the summation of its parts and the performance it can obtain. As we state in the previous point, we agree that results would be more convincing and general with reported results on natural images and other X-ray images.

---

### Decision · Program_Chairs · 2019-12-19

**Decision:**

Reject

**Comment:**

This paper proposes to use the grey level co-occurrence matrix method (GLCM) for both the performance evaluation metric and an auxiliary loss function for single image super resolution. Experiments are conducted on X-ray images of rock samples. Three reviewers provide comments. Two reviewers rated reject while one rated weak reject. The major concerns include the lack of clear and detailed description, low novelty, limited experiment on only one database, unconvincing improvement over the prior work, etc. The authors agree that the limited experiment on one database does not demonstrate the generalization capability of the proposed method. The AC agrees with the reviewers’ comments, and recommend rejection.